# Tune to Learn: How Controller Gains Shape Robot Policy Learning

*Abstract*—Position controllers have become the dominant interface for executing learned manipulation policies. Yet a critical design decision remains understudied: how should we choose controller gains for policy learning? The conventional wisdom is to select gains based on desired task compliance or stiffness. However, this logic breaks down when controllers are paired with state-conditioned policies: effective stiffness emerges from the interplay between learned reactions and control dynamics, not from gains alone. We argue that gain selection should instead be guided by learnability: how amenable different gain settings are to the learning algorithm in use. In this work, we systematically investigate how position controller gains affect three core components of modern robot learning pipelines: behavior cloning, reinforcement learning from scratch, and sim-to-real transfer. Through extensive experiments across multiple tasks and robot embodiments, we find that: (1) behavior cloning benefits from compliant and overdamped gain regimes, (2) reinforcement learning can succeed across all gain regimes given compatible hyperparameter tuning, and (3) sim-to-real transfer is harmed by stiff and overdamped gain regimes. These findings reveal that optimal gain selection depends not on the desired task behavior, but on the learning paradigm employed. Summary video can be seen in this link.

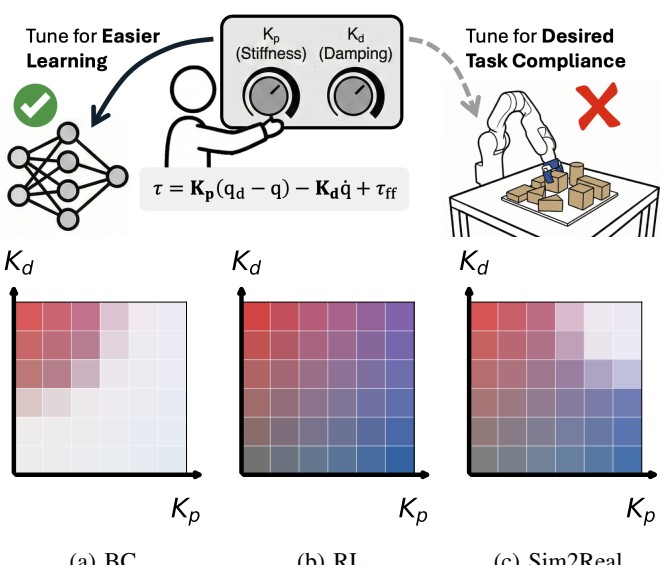

Fig. 1: **Different robot learning paradigms prefer different controller gain interfaces.** Contrary to conventional wisdom of tuning gains for desired task compliance, optimal gains depend on the learning paradigm. Based on our experimental findings, heatmaps illustrate representative gain preferences for (a) behavior cloning, which favors compliant, overdamped gains, (b) reinforcement learning, which adapts to nearly any setting, and (c) sim-to-real transfer, which is degraded by stiff and overdamped gains.

## I. INTRODUCTION

Position controllers are rapidly becoming the de facto choice for low-level control in robot learning. Their wide hardware support and intuitive nature have made them the dominant interface for executing learned manipulation policies. Yet while classical control theory provides clear guidance on selecting gains to achieve desired tracking bandwidth, disturbance rejection, or impedance characteristics, no analogous principles exist for the learning setting. An important design decision remains overlooked: how should we choose controller gains when *learning* data-driven manipulation policies?

The standard approach treats gain selection as a problem of achieving desired task behavior—contact-rich manipulation calls for low stiffness and high damping to better comply with unexpected contacts, while precision tasks call for high stiffness and low damping to accurately track position commands. But this framing conflates two distinct roles that position controllers play. When tracking open-loop trajectories, the controller *is* the *behavior*—gains directly determine how the robot responds. When paired with a learned policy, however, the controller becomes an *interface* between the policy and the physical world. The policy learns through this interface during training and acts through this interface at deployment. Viewed this way, gains function less as behavioral parameters and more as an *inductive bias*—an implicit prior over the space of closed-loop behaviors that shapes what the policy can easily express and learn.

This distinction matters because learned policies are reactive: they observe the current state and output corrective commands. A policy can achieve arbitrarily stiff or compliant task-level behavior regardless of the underlying joint-level gains, simply by modulating the magnitude and timing of its outputs. The gains, therefore, do not constrain what behaviors are reachable. We hypothesize that the gains, instead, constrain the learning problem: how easy it is to fit action labels and how errors compound during closed-loop execution, which training configurations yield successful RL policies, and whether modeling discrepancies amplify into instability during sim-to-real transfer.

Once we recognize controller gains as learning interface parameters rather than behavioral parameters, the design question becomes: which interface properties facilitate learning? And critically, do different learning paradigms prefer different interfaces, serving as a conducive *inductive bias*? We investigate these questions systematically across three paradigms of

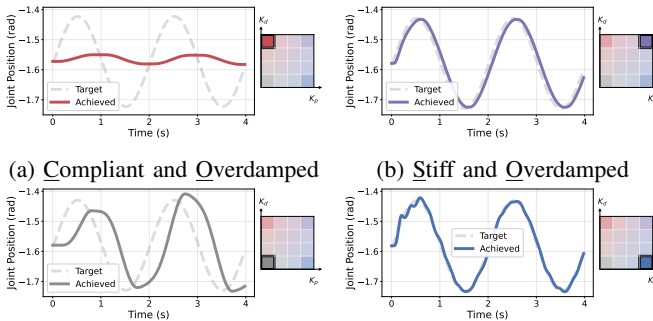

(a) Compliant and Overdamped     (b) Stiff and Overdamped

(c) Compliant and Underdamped     (d) Stiff and Underdamped

Fig. 2: **Controller gains induce diverse action–response dynamics.** We evaluate a broad range of representative gain configurations and their resulting dynamic responses to assess their impact on learnability.

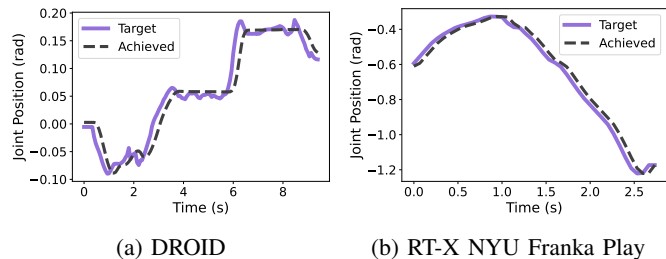

(a) DROID     (b) RT-X NYU Franka Play

Fig. 3: Tracking response curves from existing robot datasets reveal tight command-following behavior, suggesting stiff controller gains are prevalent in existing data collection pipelines.

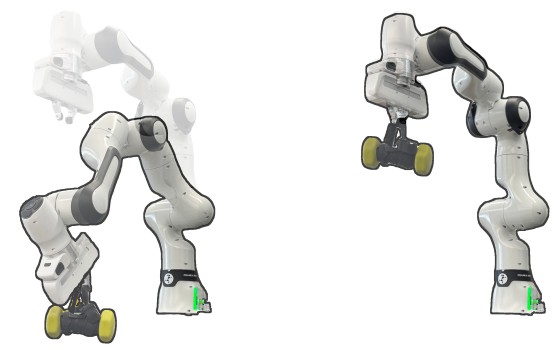

(a) Compliance w/ stiff gain     (b) Stiffness w/ compliant gain

Fig. 4: **Task-level impedance can be decoupled from low-level controller gains with learned policies.** A learned policy can achieve (a) *compliant* behavior despite stiff low-level gains, and (b) *stiff* behavior despite compliant gains.

modern robot learning and present the following findings:

1) Behavior cloning (BC) performs best with *compliant* and *overdamped* gains. Due to its inherent error dampening properties, action labels generated under compliant gain regimes (Sec. IV-A for experiment design and Sec. V-A for results)

2) Reinforcement learning (RL) from scratch is agnostic to gain setting, as long as the remaining hyperparameters are tuned to be compatible with the given gain setting. We verify this by obtaining equivalently successful RL policies for all gain regimes across multiple manipulation and locomotion tasks. (Sec. IV-B for experiment design and Sec. V-B for results)

3) When transferring policies from simulation to the real-world, *stiff* and *overdamped* controllers exacerbate the motor-level sim-to-real gap. (Sec. IV-C for experiment design and Sec. V-C for results)

Our findings converge on a unified picture of how controller gains shape learning, providing both conceptual clarity and practical guidance for this underexplored design decision.

## II. RELATED WORKS

**Position and Impedance Control.** PD control with gravity compensation [1] is the dominant low-level interface in robot learning: policies output joint position targets $\mathbf{q}_d$ tracked by $\boldsymbol{\tau} = \mathbf{K}_p(\mathbf{q}_d - \mathbf{q}) + \mathbf{K}_d(\dot{\mathbf{q}}_d - \dot{\mathbf{q}}) + \mathbf{g}(\mathbf{q})$, where the gain matrices $\mathbf{K}_p, \mathbf{K}_d$ determine joint stiffness and damping. Despite well-established stability theory [2], gain selection in practice remains largely heuristic.

**Gain Settings in Large-Scale Datasets.** Controller gain configurations in large-scale datasets are rarely documented. Analyzing DROID [3] and Open X-Embodiment [4] datasets, we find that achieved positions closely track commands with minimal lag and overshoot (Fig. 3), suggesting stiff gains have become an implicit default.

## III. DECOUPLING GAINS FROM TASK COMPLIANCE

In this section, we validate a central claim: *a closed-loop policy can realize arbitrary task-level impedance, independent of the low-level controller gains.* We demonstrate this through two counterintuitive pairings using RL policies trained to maintain a fixed pose under external disturbances, with a distance-based reward $r(\mathbf{q}) = 1 - \tanh(\|\mathbf{q} - \mathbf{g}\|^2/\lambda)$.

**Stiff behavior with compliant gains.** Despite compliant low-level gains, a small $\lambda$ encourages the policy to actively counteract disturbances (Fig. 4b).

**Compliant behavior with stiff gains.** A large $\lambda$ combined with an action smoothness penalty $\alpha\|\Delta a_t\|^2$ encourages the policy to yield smoothly under disturbances (Fig. 4a).

## IV. EXPERIMENTS

Now that we have established that gains of underlying position controllers do not dictate the task-level behavior, we aim to answer: *how do gains affect the learning process?* In this section, we present the experiment protocols we devised to study this problem systematically.

### A. Behavior Cloning (BC)

To isolate the effect of gains on learning, we need datasets where gains affect only the actions, not the state trajecto-

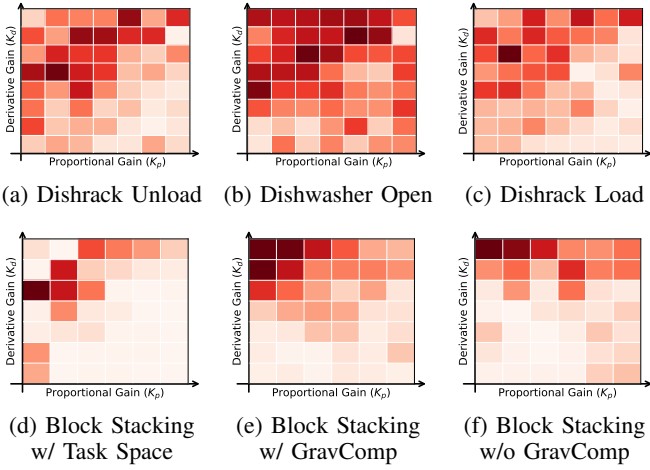

| (a) Dishrack Unload | (b) Dishwasher Open | (c) Dishrack Load |
| --- | --- | --- |
| (d) Block Stacking w/ Task Space | (e) Block Stacking w/ GravComp | (f) Block Stacking w/o GravComp |

Fig. 5: **Behavior cloning prefers *compliant* and *overdamped* controller gains.** Closed-loop rollout success rates across gain grids.

ries. Collecting demonstrations independently per gain setting confounds gain-dependent actions with different state distributions. We address this via Torque-to-Position Retargeting (TPR): we first generate demonstrations at 500Hz using torque commands, then retarget to position targets for each $(\mathbf{K}_p, \mathbf{K}_d)$:

$$\mathbf{q}_{\text{des}}(t) = \mathbf{q}(t) + \mathbf{K}_p^{-1}\left(\boldsymbol{\tau}(t) + \mathbf{K}_d\dot{\mathbf{q}}(t)\right), \qquad (1)$$

where $\tau(t), q(t), \dot{q}(t)$ are from the original torque demonstration. Retargeted commands are replayed at 50Hz, keeping only successful rollouts. This yields datasets $\mathcal{D}(s, a(\mathbf{K}))$ with nearly identical state trajectories, isolating gain-dependent actions as the sole variable. We conduct this process in simulation.

We train BC policies for each gain configuration. Our nominal setup uses a VAE with MLP, observation history length 10, and action chunk size 10, with privileged simulation states as inputs and absolute joint-space actions as outputs. We verify gain preferences are consistent across architectures (MLP vs. Transformer), model classes (regression, VAE, diffusion [5]), temporal structure, input modalities, and output representations.

### B. Reinforcement Learning

RL performance is sensitive to hyperparameters, so we must avoid conflating gain effects with suboptimal configurations. Following *environment shaping* [6], we re-tune per-joint action scales and reward weights for each gain setting via hyperparameter optimization [7]: $h^\star(\mathbf{K}) = \arg\max_h J\left(\pi^\star(h; \mathbf{K})\right)$, where $\pi^\star(h; \mathbf{K})$ is the converged policy under gains $\mathbf{K}$ and hyperparameters $h$[1].

### C. Sim-to-Real

We examine whether certain gain settings transfer more reliably from simulation to real hardware, studying reaching tasks on a Franka Research 3.

[1]We trained policies using the SKRL implementation [8] of PPO [9]. Tasks are modified from template tasks from IsaacLab [10].

**Gain-Specific System Identification.** For each gain configuration, we excite the real robot with sinusoidal targets and optimize simulation parameters $\psi$ to match state trajectories:

$$\psi^\star(\mathbf{K}) = \arg\min_\psi \sum_{t=0}^{T} \|\mathbf{x}(t; \mathbf{K}) - \bar{\mathbf{x}}(t; \psi)\|^2 \qquad (2)$$

where $\mathbf{x} = (\mathbf{q}, \dot{\mathbf{q}})$ is the real state and $\bar{\mathbf{x}}(\cdot; \psi)$ its simulated counterpart.

**Gain-Dependent Sim-to-Real Transfer.** For each gain setting, we train RL policies in the calibrated simulation, discovering transferable solutions via $h^\star(\mathbf{K}) = \arg\max_h \tilde{J}\left(\pi^\star(h; \mathbf{K})\right)$, where $\tilde{J}$ augments the objective with real-world limit penalties. Policies are deployed zero-shot. We also ablate with domain randomization (10% perturbation of system-identified parameters). We measure sim-to-real *trajectory error*:

$$\mathcal{E} = \underbrace{\|\mathbf{q}_{\text{sim}} - \mathbf{q}_{\text{real}}\|^2}_{\text{position error}} + \underbrace{\|\dot{\mathbf{q}}_{\text{sim}} - \dot{\mathbf{q}}_{\text{real}}\|^2}_{\text{velocity error}} \qquad (3)$$

averaged over 30 real-world rollouts per gain setting.

## V. RESULTS

### A. Behavior cloning

> **Result V-A-I** (Learnability): behavior cloning strongly prefers *compliant* and *overdamped* gains (i.e., top left region of Fig. 2).

Figure 5 illustrates a consistent preference for *compliant* and *overdamped* gains across a broad grid of controller settings and diverse manipulation tasks. This trend persists across training configurations: state-based vs. image-based, action-chunked vs. non-chunked, with or without state histories, task-space vs. joint-space control, and with or without gravity compensation.

**Higher MSE Loss, Better Performance.** Policies with higher closed-loop performance often exhibit *higher* validation MSE loss (Fig. 6a), indicating that compliant-regime action targets are harder to fit. Yet these policies consistently outperform their low-loss counterparts during deployment.

**Compliant Controllers Attenuate Action Errors.** The apparent paradox is explained by error-dampening properties of compliant controllers. We validate this by executing identical open-loop action sequences with injected noise across all gain configurations (Fig. 6b): compliant and overdamped gains maintain higher success rates under the same perturbations. For a given action prediction error, the robot moves *less* under compliant gains, preventing error accumulation.

### B. Online Reinforcement Learning

> **Result V-B-I** (RL Solution Existence): Online reinforcement learning *can* discover behaviors regardless of gain setpoints.

Unlike BC, on-policy RL trains on self-generated data, allowing the policy to encounter and compensate for its own

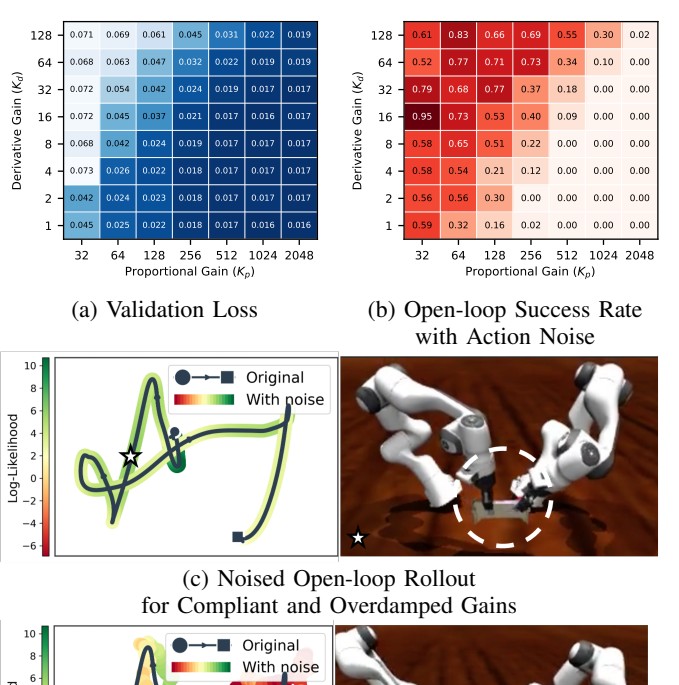

(a) Validation Loss

(b) Open-loop Success Rate with Action Noise

(c) Noised Open-loop Rollout for Compliant and Overdamped Gains

(d) Noised Open-loop Rollout for Stiff and Underdamped Gains

Fig. 6: **Compliant controllers attenuate action errors.** (a) Validation MSE loss during training: compliant gains yield higher loss, while stiff gains achieve lower loss. (b) Open-loop success rate under action noise: compliant gains maintain high success while stiff gains completely fail. (c) Compliant gains keep the perturbed trajectory close to the original, while (d) stiff gains cause large deviations that lead to task failure.

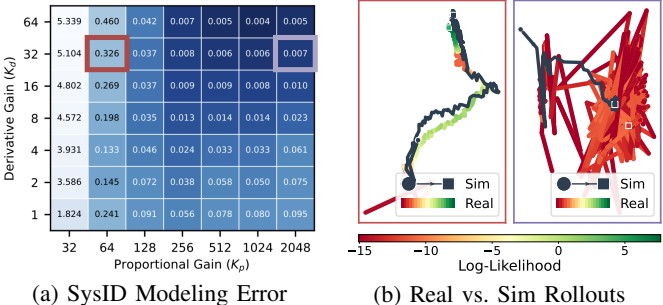

(a) SysID Modeling Error

(b) Real vs. Sim Rollouts

Fig. 7: **Stiff and overdamped gain settings yield lower SysID modeling errors, but exhibit larger closed-loop Sim2Real errors**. Policy observations during closed-loop rollout evolve similarly between sim and real (b-left) for compliant, overdamped gains, but very dissimilarly (b-right) for stiff, overdamped gains.

errors. We find that all gain regimes *can* yield working controllers given appropriate environment shaping – we verified for FR3 Joint-Reach, FR3 Lift-Cube, FR3 Open Drawer, Unitree G1 Track-Velocity, FR3 Box Reorientation, Allegro In-Hand Manipulation tasks within IsaacLab environment.

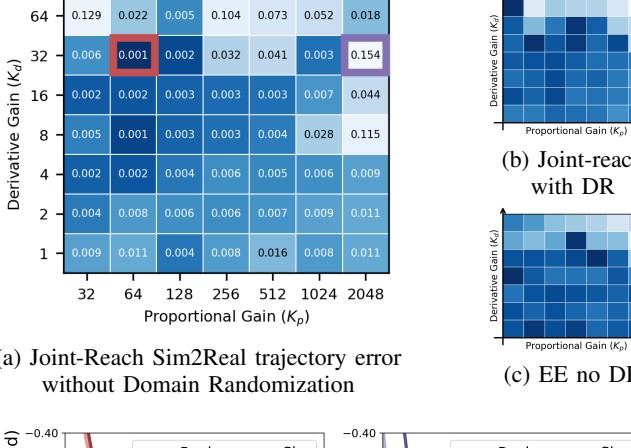

(a) Joint-Reach Sim2Real trajectory error without Domain Randomization

(b) Joint-reach with DR

(c) EE no DR

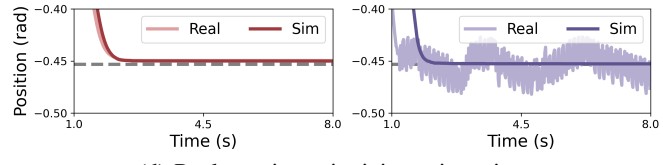

(d) Real vs. sim wrist joint trajectories.

Fig. 8: **Stiff and overdamped gain settings reduce sim2real transferability.** The Sim2Real trajectory error (Eq.3) is consistently larger (light blue) in the stiff and overdamped regime (a-c). The primary Sim2Real failure mode is high-frequency oscillation (d).

### C. Sim-to-Real

**Result V-C-I** (Sim2Real Transferability): Sim2Real transferability is lower with *stiff* and *overdamped* gain setpoints.

**Trajectory Error and Closed-Loop Amplification.** Despite yielding the lowest system identification errors (Fig. 7a), stiff and overdamped gains exhibit the worst sim-to-real trajectory error (Fig. 8). The dominant failure mode is high-frequency oscillation that persists even with domain randomization (Fig. 8b). The instability emerges not from the controller itself, but from its closed-loop interaction with the policy: stiff controllers aggressively track potentially erroneous commands, amplifying small modeling errors and pushing the policy into out-of-distribution states (Fig. 7b). This creates an inverse relationship between system identification accuracy and transfer quality—naively choosing gains that minimize modeling error can paradoxically increase sim-to-real error.

## VI. CONCLUSION AND REMARKS

We have presented a systematic study of how position controller gains shape learning dynamics across three paradigms of modern robot learning. Our findings reveal that gains function not as behavioral parameters, but as an inductive bias that modulates the learning interface between policy and environment. behavior cloning favors compliant, overdamped regimes; reinforcement learning adapts to any gain setting given compatible hyperparameters; and sim-to-real transfer suffers with stiff, overdamped configurations. These results provide both conceptual clarity and practical guidance for a widely used yet underexplored design decision.

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
