# OpenReview forum: "Tune to Learn: How Controller Gains Shape Robot Policy Learning"
_IEEE.org/ICRA/2026/Workshop/Manipulation_Robustness — ICRA 2026_

### Official Review · Reviewer_Cf3w · 2026-05-09

**Rating:** 9
**Confidence:** 4

**Review:**

Summary:
This paper investigates how position controller gains should be selected when training variable robot manipulation policies. The authors challenge using tuning gains to match desired task compliance, instead arguing gains should be chosen based on the learning paradigm in use. Since learned policies are reactive, arbitrary task-level stiffness or compliance can be achieved regardless of the underlying joint-level gains. The authors reframe gains as an inductive bias on the learning problem rather than a behavioral parameter. They show behavior cloning performs best with compliant and overdamped gains, reinforcement learning succeeds across all gain regimes given compatible hyperparameter tuning, and sim-to-real transfer is degraded by overdamped configurations.

Strengths:
The paper is well written and strongly motivated, identifying a practically important design decision that is widely used yet poorly understood across the robot learning community. The research direction is novel and broadly relevant to manipulation, with findings that generalize across tasks and robot embodiments. The Torque-to-Position Retargeting technique effectively isolated gain effects from confounding state distribution shifts, with consistent results across tasks and policy architectures. The authors show higher MSE validation loss correlates with better closed-loop BC performance.

Weakness:
The sim-to-real experiments are limited to reaching tasks on a single robot platform (Franka), so it is not immediatly clear if the findings generalize for contact-rich manipulation or other hardware. The paper does not explore alternative sim-to-real methods beyond system identification with optional domain randomization, and techniques such as adaptive domain randomization or dynamics-aware policy training are not considered. The work also does not address joint training paradigms such as vision-language-action models (VLAs) combined with online fine-tuning, which are increasingly common in practice and may interact with gain selection in non-trivial ways, but perhaps this is out of scope for this paper.

Overall, a very interesting paper well fitted for this workshop.

---

### Official Review · Reviewer_PQUB · 2026-05-16
**A good paper topic and a meaningful empirical study, but the paper currently overclaims generality.**

**Rating:** 7
**Confidence:** 4

**Review:**

**Summary**:

The paper studies how low-level PD position-controller gains (Kp,Kd) affect robot policy learning. Its main claim is that gains should be chosen for learnability, not just desired compliance/stiffness: bc prefers compliant, overdamped gains; rl can succeed across almost all gain regimes if hyperparameters are retuned; and sim2real transfer is worse for stiff, overdamped gains even when those gains give lower system-identification error. The paper supports this with a decoupling argument, bc experiments with torque-to-position retargeting, rl experiments with per-gain hyperparameter optimization, and a sim-2-real study on a franka robot.

**Weaknesses**:

1. The 3 conclusions from the paper feels almost too clean relative to the actual data. Real systems are likely a lot more messier. Different tasks, policy classes, action parameterizations, control rates, delay/noise levels, and hardware unpredictability may move these boundaries substantially. The paper’s conclusions feel more like trends under the chosen setup than robust generic recipes.

2. The paper does not clearly distinguish performance, sample efficiency, and tuning burden. For rl, the authors show that successful solutions exist across gain regimes after per-gain retuning. But that does not mean gains do not matter for learning. Some gains may require much more careful tuning, more interactions, or lower robustness. The paper’s formulation makes it easy to miss this because it focuses on eventual success after optimization, not learning efficiency under the same amount of tuning effort.

3. The bc result may be more about deployment robustness than learnability per se. One of the paper’s most interesting findings is that higher validation mse can correspond to better closed-loop performance. But that actually undermines the paper’s claim about action-label fitting: the better-performing controllers are harder to fit in action space, yet more “forgiving” at execution time because they attenuate errors made by less accurate prediction. That sounds less like “bc learns better” and more like “the control interface masks predictor errors better.” I feel this distinction is not made clear enough.

---

### Decision · Program_Chairs · 2026-05-21

Accept